# Diet Quality Indices and Physical Activity Levels Associated with Adequacy of Gestational Weight Gain in Pregnant Women with Gestational Diabetes Mellitus

**DOI:** 10.3390/nu13061842

**Published:** 2021-05-28

**Authors:** Vanessa Averof Honorato de Almeida, Rafaela Alkmin da Costa, Cristiane de Freitas Paganoti, Fernanda Cristina Mikami, Ana Maria da Silva Sousa, Stela Verzinhasse Peres, Marco Antonio Borges Lopes, Rossana Pulcineli Vieira Francisco

**Affiliations:** Disciplina de Obstetrícia, Departamento de Obstetrícia e Ginecologia, Faculdade de Medicina FMUSP, Universidade de São Paulo, Av. Dr. Enéas Carvalho de Aguiar, 255, Cerqueira César 05403-000, São Paulo, Brazil; vanessa.averof@fm.usp.br (V.A.H.d.A.); rafaela.alkmin@hc.fm.usp.br (R.A.d.C.); c.paganotti@hc.fm.usp.br (C.d.F.P.); fernanda.cristina@hc.fm.usp.br (F.C.M.); ana.mssousa@hc.fm.usp.br (A.M.d.S.S.); svperes80@gmail.com (S.V.P.); marco.lopes@hc.fm.usp.br (M.A.B.L.)

**Keywords:** gestation, gestational diabetes, diet, physical activity, gestational weight gain

## Abstract

The aim of this study was to evaluate the Diet Quality Index (DQI) and the Physical Activity (PA) levels associated with adequacy of gestational weight gain in pregnant women with gestational diabetes mellitus (GDM). A total of 172 pregnant women with a single fetus and a diagnosis of GDM participated. Food intake was self-reported on the food frequency questionnaire and DQI was quantified using the index validated and revised for Brazil (DQI-R). To assess PA, the Pregnancy Physical Activity Questionnaire was administered. Gestational weight gain was classified, following the criteria of the Institute of Medicine, into adequate (AWG), insufficient (IWG), or excessive (EWG) weight gain. A multinomial logistic regression analysis was performed, with level of significance <0.05. The participants were divided into 3 groups: AWG (33.1%), IWG (27.3%), and EWG (39.5%). The analysis indicated that if the pregnant women PA fell into tertile 1 or 2, then they had a greater chance of having IWG, whereas those with the lowest scores on the DQI-R, whose PA fell into tertile 2, and pregestational obesity women had the greatest chance of having EWG. This study has shown that low PA levels may contribute towards IWG. On the other hand, a low final DQI-R score, representing inadequate food habits, low PA levels, and pregestational obesity may increase the chance of EWG in patients with GDM.

## 1. Introduction

Malnutrition and the growing problem of excess weight and obesity in women of reproductive age are associated with several maternal and fetal adverse outcomes. One such complication is gestational diabetes mellitus (GDM), which, in the long run, is related to a greater postpartum weight retention, complications in a future pregnancy, and an increased risk of type 2 diabetes mellitus (DM2) for the mother [1,2,3,4]. The prevalence of hyperglycemia, according to the diagnostic criteria of the World Health Organization (WHO), is estimated at 16.6% during pregnancy, and GDM represents 84% of these cases [5,6].

Nutrition during pregnancy plays an important role in adequate weight gain, the management of glycemic control, and fetal and neonatal outcomes. However, several gaps in knowledge remain as to the influence of pregestational diet quality and physical activities (PAs) on maternal and perinatal outcomes [7,8,9]. Gestational weight gain (GWG) is an important health and quality of life indicator in the life of women and their fetuses. Excessive or insufficient weight gain requires attention in the prenatal care of all women, especially those with GDM, for it may influence glycemic control and, consequently, the pregnancy outcome and postpartum weight retention [10]. Thus, the pregestational nutritional status and the monitoring of weight gain during pregnancy are actions that should actually and routinely be carried out in prenatal care aiming at reduction in maternal and fetal risks [11]. Therefore, the objective of this study was to evaluate the association between diet quality indices (DQIs), along with PA levels and weight gain adequacy, during the pregnancy of women with GDM.

## 2. Materials and Methods

### 2.1. Study Population

This was a descriptive prospective cohort study involving pregnant women with a diagnosis of GDM (NCT 03307486).

The study included pregnant women aged 18 years or older with a single fetus, a diagnosis of GDM according to the International Association of Diabetes and Pregnancy Study Groups [12] (initial fasting glucose ≥92 mg/dL and <126 mg/dL or 75 g oral glucose tolerance test with fasting ≥92 mg/dL and/or after 1 h ≥ 180 mg/dL and/or after 2 h ≥ 153 mg/dL), absence of glucose intolerance prior to pregnancy (defined by previous diagnosis of polycystic ovary syndrome or by fasting glucose ≥100 mg/dL or 2 h after overload with 75 g of glucose ≥140 mg/dL or glycated hemoglobin ≥5.7% before pregnancy), nonchronic use of glucocorticoids or antiretroviral drugs for HIV viruses given their diabetogenic effect and potential confounding factor, and a signed free and informed consent statement (FICS). Excluded were the pregnant women who could not understand and/or respond to the research questionnaire items, who missed the prenatal care visits, and whose medical records were incomplete.

The pregnant women with a diagnosis of GDM [12] were followed up at the clinic in accordance with the current assistance protocol. After the diagnosis of GDM, all of the women were invited to participate in a multidisciplinary group to receive guidance on diet, PA, and glucose monitoring. If the treatment goal (fasting glucose ≤95 mg/dL and 1 h postprandial ≤140 mg/dL) was not reached with lifestyle changes, insulin therapy was indicated. Before joining the multidisciplinary group, they were invited to participate in the research protocol of this study. Those who agreed to participate received all the information concerning the study. Only after reading and signing the FICS were they asked about their pregestational weight, food consumption, and PA on the preestablished structured questionnaires on paper [13].

This research was approved by the Ethics Committee for Project Analysis of the Clinical Board of the Hospital das Clínicas da Faculdade de Medicina da Universidade de São Paulo HC-FMUSP (CAPPesq), CAAE No. 48868915.9.0000.0068.

Clinical and demographic data were obtained by accessing the patients’ physical and electronic medical records. The pregestational weight mentioned above was used to estimate weight gain during pregnancy and pregestational body mass index (BMI). The pregestational BMI was categorized as low weight (<18.5 kg/m^2^), adequate weight (18.5–24.9 kg/m^2^), overweight (25.0–29.9 kg/m^2^), and obesity (≥30.0 kg/m^2^) [14]. The pregnant women’s weight was measured in every medical consultation; however, in this study, we used only the last weight measurement, which was made at the time of delivery, to calculate the total weight gain during pregnancy. The main outcome to be assessed was maternal weight gain, as recommended by the Institute of Medicine (IOM) [14], and it was classified into adequate weight gain (AWG), insufficient weight gain (IWG), and excessive weight gain (EWG).

### 2.2. Dietary Intake Assessment and Development of Diet Quality Scores

Food consumption was collected by the quantitative food frequency questionnaire (FFQ) of 101 items, which was validated to assess the usual food intake by individuals aged 18 years or older. The FFQ is based on a list of the most consumed foods by the Brazilian population, and it estimates the average consumption of the previous 12 months [13]. It took approximately 20 min to administer the questionnaire to each participant. Diet quality was evaluated using the diet quality index, revised and validated for the Brazilian population (DQI-R) [15]. The DQI-R consists of 12 components characterizing the different aspects of a diet, and they add up to a maximum of 100 points, a score equivalent to a high-quality diet. The components may be categorized as adequacy components, in which case increased intake is recommended, or as moderation components, in which case restricted intake is recommended. There are 9 adequacy components (“total cereals,” “whole grains,” “total fruits,” “whole fruits,” “total vegetables,” “dark green vegetables and oranges and legumes,” “milk and dairy products,” “beef, eggs, and legumes,” and “oils”) and 3 moderation components (“saturated fat,” “sodium,” and “AA fats”; the energy of the latter is provided by solid fat, added sugar, and alcohol). The lowest score is zero. The maximum score varies according to the component as follows: 5 for the first six components, 10 for the next five components, and 20 for the last component. The questionnaire scores were subclassified into tertiles to enable group comparison.

### 2.3. Physical Activity Assessment

PA was assessed by means of the Pregnancy Physical Activity Questionnaire [16], validated for and adapted to the Portuguese language. This tool captures the daily time spent on PAs in the previous three months as follows: (1) during leisure and while practicing physical exercises and sports; (2) at work and as means of communication; and (3) involved in taking care of other people and doing housework, for example. The questionnaires took approximately 10 min per participant to be evaluated. Each woman was instructed individually on how to answer the questionnaire, and thus, became responsible for its completion. This procedure was adopted to avoid the researcher’s influence on each patient’s responses. The questionnaire scores were subclassified into tertiles for group comparison.

### 2.4. Statistical Analysis

To analyze the categorical variables, the chi-square test or the Fisher test was used, as appropriate. If data distribution was not normal, the quantitative variables were compared with the Kruskal–Wallis test. If a variable was statistically different, Dunn’s post hoc test was used. The statistical analyses were performed with SPSS version 23.0 (SPSS Inc., Chicago, IL, USA), with significance level set at <0.05.

According to the main outcome, a univariate analysis was carried out to compare the patients in terms of the following: maternal age, schooling, marital status, color, work status, family history of diabetes, previous GDM, smoking, parity, hypertension, test used for diagnosing GDM, BMI at the time of the GDM diagnosis, insulin need for controlling metabolism, BMI in late pregnancy, tertile of the final score on the DQI-R, and tertile of the total PA score.

For multiple multinomial logistic regression analysis, the clinically significant variables and those with *p* < 0.20 in the univariate analysis were tested by the Stepwise technique. The evaluation of the model’s quality followed the parameters, namely: alteration of β values < 10%, maintenance or reduction of 95% CI, and 10 outcomes for each 1 degree of freedom [17].

## 3. Results

A total of 226 women with a single pregnancy and a GDM diagnosis was recruited between May 2017 and October 2018. As shown in Figure 1, fifty-four women were excluded. The remaining 172 were divided into three groups, according to their adequacy of weight gain during pregnancy (calculated from their pregestational BMI), as recommended by the IOM [14]. The three categories were the following: IWG, 47 (27.3%); AWG, 57 (33.1%); and EWG, 68 (39.5%).

The pregnant women’s features can be seen in Table 1. There were statistically significant differences between the groups in terms of schooling and work status. The women with EWG took insulin more frequently (36.8%) during pregnancy (*p* = 0.001). The groups did not differ with respect to pregestational BMI (Table 1). However, there were significant differences between the groups relative to the BMI classification at the end of pregnancy (Table 1). Gestational age (GA) at GDM diagnosis (EWG 16.30 ± 8.42 vs. AWG 16.24 ± 8.89 and IWG 17.68 ± 8.18; *p* = 0.647), GA at the first evaluation of GDM (EWG 23.16 ± 7.33 vs. AWG 23.21 ± 7.83 and IWG 23.14 ± 6.89; *p* = 0.982), and GA at delivery (EWG 38.70 ± 1.11 vs. AWG 38.03 ± 1.88 and IWG 38.48 ± 1.27; *p* = 0.093) did not differ statistically between the groups.

There was a significant difference between the groups in terms of weekly weight gain during pregnancy. The weight gain median per week was increased more in the EWG women (0.413 g/week) as opposed to the AWG women (0.222 g/week) and to the IWG women (0.118 g/week), *p* < 0.001.

The classification of the pregnant women according to the tertiles of the final DQI-R scores of the Brazilian population [15] show that most of them are in tertile 1 (27.6–50) (Table 2).

When examined separately, the variables of the adequacy components of the DQI-R, according to the classification of weight gain during pregnancy, show that most of the pregnant women in the three groups received low scores in whole grains (AWG: 0.23; IWG: 0.09; EWG: 0.11; *p* = 0.650), in oils (AGW: 2.85; IGW: 2.70; EWG: 2.75; *p* = 0.600) and in milk, dairy products, and soy beverages (AGW: 4.60; IWG: 3.82; EWG: 5.27; *p* = 0.220), indicating low consumption of such foods in the three groups; there was no statistical difference between the groups.

Examination of the variables of moderation components of the DQI-R showed that most pregnant women in the three groups had low sodium (AWG: 4.22; IWG: 3.50; EWG: 3.78; *p* = 0.145) and saturated fat (AWG: 4.59; IWG: 4.56; EWG: 3.80; *p* = 0.493) scores, indicating a high consumption of these components in all groups; there was no statistical difference between the groups.

There was a significant association between the final PA score and adequacy of weight gain during pregnancy (*p* = 0.020). Of the pregnant women with AWG, 49.1% were in the last tertile of the PA score (>135.35–353.25 METs/week) (Table 2).

The multiple multinomial regression with the moderation components of the DQI-R, the final score of the DQI-R, the total PA score, and the pregestational BMI identified the PA score as an independent factor associated with IWG. The pregnant women with GDM who practiced PAs and were in tertile 1 (≤84 METs/week) stood a 3.80-fold chance of IWG (CI 95%: 1.32–10.92; *p* = 0.013), and those who practiced PAs and were in tertile 2 (>84–135.50 METs/week) had a 3.99-fold greater chance of having IWG (CI 95%: 1.34–11.90; *p* = 0.013) than pregnant women with PA in tertile 3 (>135.35–353.25 METs/week) (Table 3).

The factors of independent association of the pregnant women with GDM who presented EWG were the final DQI-R score, the total PA score, and the pregestational BMI. The pregnant women whose final DQI-R score fell in tertile 1 (27.6–50), indicating an inadequate diet, had a 2.33 times greater chance of EWG (CI 95%: 1.02–5.36; *p* = 0.046) than those with the DQI-R in tertile 2. Those with PA in tertile 2 (>84–135.50 METs/week) had a 3.47 times greater chance of EWG (CI 95%: 1.36–8.89; *p* = 0.009) than those whose PA was in tertile 1. The pregnant women with pregestational BMI classified into the obesity group stood a 3.20-fold greater chance of EWG (OR: 3.20; CI 95%: 1.14–8.99; *p* = 0.027) than those with an adequate pregestational BMI (Table 3).

Of the EWG patients, 36.8% took insulin, and of the AWG patients, 12.3% (X^2^ = 9.759; *p* = 0.002) did so. Considering that insulin use was more frequent in the EWG group, an estimate was made for the underlying reason for such a difference according to the classification of gain weight during pregnancy. The result was that EWG patients exhibited an OR equal to 4.15 (CI 95%: 1.83–12.07) for insulin use compared to the pregnant women with AWG.

## 4. Discussion

This study has enabled the conclusion that low PA levels may contribute to IWG. On the other hand, a low score on the DQI-R, which translates into inadequate food habits, associated with low PA scores and a classification of pregestational obesity, may contribute to EWG.

A woman’s weight at the beginning of pregnancy may have a great impact on maternal and fetal health [10,18]. The high percentage of obesity (36.6%) and overweight (36%) and not a single case of low weight observed among the pregnant women in this study reflect an excessive weight increase gain among the women of reproductive age in Brazil [19] and worldwide [20,21]. This result is consistent with the findings in the literature, which point to the fact that the majority of pregnant women with GDM are in the obesity or overweight category when they get pregnant [18,22]. According to Hernández-Higareda et al. (2017) [23], pregestational obesity has been identified as a risk factor with a nearly twofold chance for GDM (OR: 1.95; CI 95%: 1.39–2.76; *p* < 0.001).

In our results, analysis of the multinomial regression showed that pregestational obesity increased the risk for EWG threefold. A cohort study conducted at the Hospital of the University of North Carolina, which included pregnant women diagnosed with GDM, demonstrated that women with obesity gained less weight during pregnancy, but were more prone than normal-weight women to exceed the IOM guidelines for total weight gain [22], as was the case in our study population.

In our study, 39.5% of the pregnant women gained excessive weight. Viecceli et al. (2015) [3], in their meta-analysis, concluded that AWG occurs in only a third of the pregnancy cases with GDM. The EWG was frequent and led to a larger number of adverse outcomes. The authors suggest that smaller gains would be beneficial, except to women with an adequate pregestational BMI. Therefore, an effective prevention of EWG, as currently defined by the IOM and based on pregestational BMI, is extremely important for patients with GDM.

A retrospective study published in 2018 by Komem et al. [24], which aimed at evaluating the association between gestational weight gain or loss and an adverse outcome of pregnancy in women with GDM, showed that IWG and monitored weight loss may be associated with better maternal and neonatal outcomes.

Furthermore, some authors wonder if the recommendation to gain weight during pregnancy proposed by the IOM may be extended to pregnancies with GDM. They suggest more limited objectives than those of the IOM to reduce the prevalence of adverse pregnancy outcomes primarily in women with obesity and GDM [25,26,27].

Our DQI-R results demonstrate that the food habits of the study population are poor and should be improved. Most of the pregnant women’s (66.9) diet quality was classified as “inadequate diet,” showing the need to address nutrition education specifically for women of reproductive age. After the adjustments of the multinomial regression, a statistically significant association was found between a lower DQI-R score and a higher chance of EWG. Our findings corroborate those of Gadgil et al. 2019 [28], whose purpose was to investigate the association between diet quality and glycemic control in women with GDM. In their study, the diet quality of most pregnant women was low, and the higher the final score, the higher the diet quality and the better the overall and the postprandial glycemic control among women with GDM.

Parker et al. 2019 [29], in a study (Infant Feeding Practices Study II) assessing the relationships between pregestational BMI, total gestational weight gain, and pregestational DQI, observed that the relationship between gestational weight gain and pregestational DQI was dependent on pregestational BMI. In other words, women in the overweight and in the obesity categories prior to pregnancy were at a higher risk of a low pregestational DQI score, which represents low diet quality and higher risk of EWG.

Analysis of the total PA of our population reveals that the maximum energy expenditure reached 353.25 METs/week, below the recommended PA value for staying healthy. According to Haskell et al. [30], for adults to maintain their health, their individual energy expenditure should lie between 450 to 750 METs/week after adding up all of the physical activities undertaken, be they sports or daily movements. The specific recommendations for physical activity throughout pregnancy should equal at least 150 min of moderate-intensity PAs per week (which corresponds to 450 to 750 METs/week), to achieve clinically meaningful health benefits and reductions in pregnancy complications [31].

The present study has shown that women who do not engage in adequate PAs run a higher risk of IWG or EWG, leading to the realization that physical inactivity remains a significant public health concern. The PA results of the 2019 National Health Research (Pesquisa Nacional de Saúde (PNS)), carried out by the Brazilian Institute of Geography and Statistics (Instituto Brasileiro de Geografia e Estatística (IBGE)) partnering with the Health Ministry (Ministério da Saúde), showed that in the Brazilian adult population, 40.3% of the individuals were classified as insufficiently active, that is, they did not undertake PA or limited it to not even 150 min a week. In all of the large regions, there was a higher proportion of insufficiently active women than men [32].

Studies report that there is a greater chance of detecting GDM among women who engage in light to moderate PA [33,34].

In the present study, a significant association was found between the final PA score and adequacy of total gestational weight gain in patients with GDM (*p* = 0.020). The association with the distribution of the tertiles of PA levels also produced significant results when activities were light (*p* = 0.012) or moderate (*p* = 0.024).

Gou et al. (2019) [35] conducted a retrospective study involving 1523 Chinese women, aiming to evaluate the association between pregnancy weight gain and pregnancy outcomes in women with GDM. In total, 451 (29.6%) women presented IWG, and 484 (31.8%), EWG. A restrictive diet and excessive physical activity accounted for the IWG and were associated with a higher rate of preterm births. The authors highlighted the fact that the extremes of weight gain in pregnancy (IWG and EWG) should be carefully watched in women with GDM, and they stressed that nutritional therapy and physical exercises should be encouraged to maintain glycemic control and adequate weight gain during pregnancy, thus reducing the chance of adverse outcomes.

It is worth noting that, in the present study, the association between EWG and insulin use was significant; hence, women with EWG had a 4.15-fold greater chance (CI 95%: 1.63–10.55; *p* = 0.003) of using insulin during a pregnancy with GDM than pregnant women with AWG. This result is in agreement with the data obtained by Barnes et al. 2020 [36]. These authors, after the adjustments for confounding factors, noted a 1.4 times greater chance of insulin therapy in pregnant women with EWG (CI 95%: 1.1–1.7; *p* < 0.01). For each 2-kg increment in weight gain, the chance of insulin therapy increased 1.3-fold (CI 95%: 1.1–1.5; *p* < 0.001).

High-quality data indicate that diet and/or physical exercises during pregnancy may reduce the risk of excessive weight gain. Such interventions are an important part of the care needed for gaining adequate weight in pregnancy; nevertheless, further research is necessary to establish safe guidelines [4].

This study has a few limitations. A comprehensive dietary assessment was concluded soon after the GDM diagnosis and there were no evaluations following the instructions the pregnant women received for the GDM treatment. Analysis established a temporal relationship between DQI-R, PA levels, and adequacy of weight gain during pregnancy. This suggests that PA levels and diet quality during the months prior to GDM diagnosis may have persistent effects on weigh control and adverse effects during a pregnancy with GDM.

The initial report on the food consumption of all participants may reflect a recall bias toward the usual intake or a weakness of the PAQ itself. Such a limitation may weaken the association between the components of the DQI-R score and inadequate weight gain during pregnancy.

Besides, the studies investigating the relationship between diet quality and food habits before and after the pregnancy of women with GDM are scarce and were conducted in regions with food habits that differ from those of the Brazilian culture, thus limiting a comparison of the findings [37,38,39,40].

It is necessary to emphasize that, despite the limitations, this study has great potential, for it evaluates an issue of extreme clinical relevance, namely adequacy of gestational weight gain, by means of low-cost tools with a high descriptive potential, which are the food habits and PA validated questionnaires. These aim to identify intervention targets, which may bring a positive impact on pregnancy; however, this is a subject matter still undefined in the literature.

## 5. Conclusions

This study has concluded that low PA levels may contribute to IWG, while a low DQI-R score, which is equivalent to inadequate food habits; low PA levels; and pregestational obesity are associated with EWG.

Most of the pregnant women were found to have a low DQI-R score and a low PA level. The study results strengthen the idea that prevention of obesity and excess weight, as well as guidance on diet quality and PAs, is extremely necessary for public health on a global scale. Furthermore, additional research should be carried out to clarify the relationship between diet quality and eating patterns before and during pregnancy in the Brazilian and non-Brazilian populations.

## Figures and Tables

**Figure 1 nutrients-13-01842-f001:**
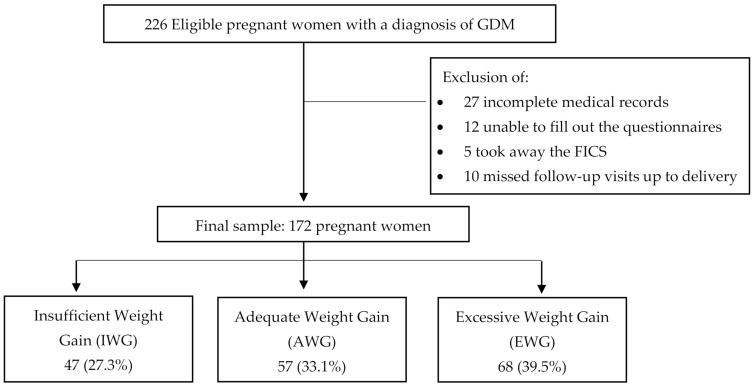
Patient selection and group allocation according to adequacy of weight gain during pregnancy.

**Table 1 nutrients-13-01842-t001:** Maternal characteristics.

Gestational Weight Gain				
	(AWG)	(IWG)	(GPE)	
	*n* = 57	*n* = 47	*n* = 68	
Variables	*n* (%)	*n* (%)	*n* (%)	*p* Value
Relationship status				
With partner	29/45 (64.4)	25/40 (62.5)	30/59 (50.85)	0.311
Color				
White	47 (82.46)	36 (76.60)	56 (82.35)	0.690
Nonwhite	10 (17.54)	11 (23.40)	12 (17.65)	
Work status	45 (78.95)	23 (48.94)	47 (69.12)	**0.005**
Hypertension	14 (24.6)	12 (25.5)	22 (32.4)	0.572
Smoking	3 (5.3)	2 (4.3)	4 (5.9)	1.000
Family history of DM	38 (66.7)	25 (53.2)	40 (58.8)	0.368
Personal history of GDM	5 (8.8)	4 (8.5)	7 (10.3)	1.000
Primigravida	16 (28.1)	10 (21.3)	19 (27.9)	0.670
GDM diagnosis				
OGTT 75 g	22 (38.6)	23 (48.9)	25 (36.8)	
Fasting blood glucose	35 (61.4)	24 (51.1)	43 (63.2)	0.394
Insulin use in pregnancy	7 (12.3)	6 (12.8)	25 (36.8)	**0.001**
Pregestational BMI				
Adequate	16 (28.1)	18 (38.3)	13 (19.1)	
Overweight	25 (43.8)	11 (23.4)	26 (38.2)	0.068
Obesity	16 (28.1)	18 (38.3)	29 (42.6)	
Atalah BMI at the end of pregnancy with GDM				
Low weight	4 (7.0)	4 (8.5)	0 (0.0)	
Adequate	10 (17.5)	17 (36.2)	3 (4.4)	
Overweight	25 (43.9)	10 (21.3)	15 (22.1)	**˂0.001**
Obesity	18 (31.6)	16 (34.0)	50 (73.5)	

Abbreviations: AWG, adequate weight gain; IWG, insufficient weight gain; EWG, excessive weight gain; DM, diabetes mellitus; GDM, gestational diabetes mellitus; OGTT-75 g, oral glucose tolerance test with a 75-g overload; BMI, body mass index, Bold *p*-value <0.05.

**Table 2 nutrients-13-01842-t002:** Revised Diet Quality Indices and of the final PA scores of pregnant women with GDM according to gestational weight gain.

Gestational Weight Gain				
	(AWG)	(IWG)	(EWG)	
	*n* = 57	*n* = 47	*n* = 68	
Variables	*n* (%)	*n* (%)	*n* (%)	*p* Value
Tertiles of the Final DQI-R Scores (Score)				
1st Tertile (27.6–50)	35 (61.4)	29 (61.7)	51 (75.0)	
2ndTertile (>50–60)	22 (38.6)	18 (38.3)	16 (23.5)	0.163
3rd Tertile (>60–80)	0 (0.0)	0 (0.0)	1 (1.5)	
Final PA Score (METs/week)				
1st PA Tertile (≤84)	16 (28.1)	20 (42.6)	21 (30.9)	
2nd PA Tertile (>84–135.50)	13 (22.8)	18 (38.3)	26 (38.2)	**0.020**
3rd PA Tertile (>135.35–353.25)	28 (49.1)	9 (19.1)	21 (30.9)	

Abbreviations: AWG, adequate weight gain; IWG, insufficient weight gain; EWG, excessive weight gain; DQI-R, Diet Quality Index-Revised; PA, physical activity; MET, metabolic equivalent of task; Bold *p*-value <0.05.

**Table 3 nutrients-13-01842-t003:** Multiple multinomial regression analysis for weight gain recommendation to pregnant women with GDM.

Weight Gain Recommended during Pregnancy ^a^		OR	CI 95%		*P*
			Inferior	Superior	
Insufficient	Sodium (0–10)	0.83	0.66	1.04	0.098
	Saturated Fats (0–10)	1.05	0.90	1.23	0.503
	AA Fat (0–20)	1.03	0.93	1.14	0.547
	Final DQI–R Score (27.6–50)	1.07	0.45	2.55	0.879
	Final DQI–R Score (>50–60) *	1.0			
	1st PA Tertile (≤84 METs/week)	3.80	1.32	10.92	***0.013***
	2nd PA Tertile (>84–135.50 METs/week)	3.99	1.34	11.90	***0.013***
	3rd PA Tertile (>135.35–353.25 METs/week)	1.0			
	Pregestational BMI (Overweight)	0.49	0.17	1.38	0.177
	Pregestational BMI (Obesity)	1.29	0.46	3.61	0.631
	Pregestational BMI (Adequate)	1.0			
Excessive	Sodium (0–10)	0.91	0.74	1.12	0.383
	Saturated Fats (0–10)	0.92	0.80	1.07	0.270
	AA Fat (0–20)	1.08	0.98	1.18	0.106
	Final DQI-R Score (27.6–50)	2.33	1.02	5.36	***0.046***
	Final DQI-R Score (>50–60) *	1.0			
	1st PA Tertile (≤84 METs/week)	1.95	0.77	4.95	0.162
	2nd PA Tertile (>84–135.50 METs/week)	3.47	1.36	8.89	***0.009***
	3rd PA Tertile (>135.35–353.25 METs/week)	1.0			
	Pregestational BMI (Overweight)	1.79	0.67	4.79	0.249
	Pregestational BMI (Obesity)	3.20	1.14	8.99	***0.027***
	Pregestational BMI (Adequate)	1.0			

**^a^** The reference category is adequate. ***** An individual scored above 80 and was placed in the 50-to-60 category. Abbreviations: AA fat, energy sourced from solid fat, alcohol, and added sugar; DQI-R, Diet Quality Index-Revised; PA, physical activity; MET, metabolic equivalent of task; BMI, body mass index; Bold *p*-value < 0.05.

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
