# Peer review of "Diet Quality Indices and Physical Activity Levels Associated with Adequacy of Gestational Weight Gain in Pregnant Women with Gestational Diabetes Mellitus"

_nutrients, 2021, doi:10.3390/nu13061842_

Round 1

Reviewer 1 Report

The purpose of this paper was to examine diet quality and physical activity levels and their association with adequate gestational weight gain in pregnant women with gestational diabetes mellitus. The majority of women had adequate or insufficient weight gain. Low levels of physical activity may contribute to insufficient weight gain but poor diet quality, low physical activity, and prepregnancy obesity may increase the risk of excessive gestational weight gain. These findings are important to the existing evidence of gestational weight gain and gestational diabetes as efforts to help manage gestational weight gain and gestational diabetes is a major public health concern.

Broad comments:

The article was well written and the points were clear and concise.

The abstract could use a broad impact statement.

There were some spacing issues in the abstract and in the introduction when the article was printed.

Please make sure abbreviations are consistent across the body of the manuscript, tables, and figures.

Use person first language – for example, should be referring to pregnant women with gestational diabetes, pregnant women with overweight and/or obesity NOT the overweight and/or obese.

Specific comments:

Line 17: There is no space between the words and the abbreviations

Line 22-24: This sentence should be split into two. It reads strangely and is difficult to follow.

Line 43-46: This point is important but could cause confusion as to what “diagnosis of pregestational nutrition” is. This could use clarification and elaboration.

Materials & Methods

Pre-pregnancy weight was self-reported correct? If data has been extracted from the medical record, why not use that pre-pregnancy weight measurement?

Were the surveys completed at the doctor’s office during the prenatal visit? Were they conducted on paper or online? A bit more explanation of these procedures would be appreciated.

Figure 1: There are two adequate weight gain boxes – should one be insufficient weight gain?

What was the mean gestational age? Was there a cut off for gestational age and recruitment?

GDM diagnosis timing would be an interesting statistic to know as well.

Since there was a small percentage of primiparous women, did GWG differ between primi and others? These results would be interesting especially if there was a higher incidence of GDM in women with previous pregnancies.

Lines 138 and 165: Are these the correct abbreviations? In the abstract, adequate weight gain is AWG and insufficient weight gain is IWG and these are how they are labeled in the tables.

Line 185: Bold MET

Insulin treatment seemed to be the main focus of these results…were there other treatment options women? It would be interesting to see how these treatment options influenced dietary patterns and physical activity especially since those two are main treatment options for GDM.

Line 246: Are these pregnant specific recommendations or for the general public?

The paragraphs starting at line 244-248 and 249-257 – can you connect these points better? The first paragraph talks about METs/week but the second paragraph talks about 150 min/week. Having those connect on one type of outcome will create a stronger point and be more cohesive.

Author Response

May 13, 2021

Dear Reviewer,

Thank you for your review. We accepted your corrections explaining point-by-point the details of the revision and made the changes as requested.

Broad comments:

The article was well written and the points were clear and concise. The abstract could use a broad impact statement.

Reply: Thanks

There were some spacing issues in the abstract and in the introduction when the article was printed.

Reply: Thanks, they were corrected.

Please make sure abbreviations are consistent across the body of the manuscript, tables, and figures.

Reply: The manuscript was reviewed and the corrections were made.

Use person first language – for example, should be referring to pregnant women with gestational diabetes, pregnant women with overweight and/or obesity NOT the overweight and/or obese.

Reply: Corrections were made in lines 200, 223, 234, 230, 247, 261, and 262.

Specific comments:

Line 17: There is no space between the words and the abbreviations

Reply: Thanks, spacing was corrected.

Line 22-24: This sentence should be split into two. It reads strangely and is difficult to follow.

Reply: Thanks, it has been changed.

Exclusion of lines 22- 24: Low PA levels may contribute to IWG, while low DQI-R indices, which represent inadequate food intake, low PA levels, and a classification of pregestational obesity possibly increase the chance of having EWG in patients with GDM.

Inclusion of lines 22 -24: “This study has shown that low PA levels may contribute towards IWG. On the other hand, a low final DQI-R score, representing inadequate food habits, low PA levels, and pregestational obesity may increase the chance of EWG in patients with GDM.”

Line 43-46: This point is important but could cause confusion as to what “diagnosis of pregestational nutrition” is. This could use clarification and elaboration.

Reply: Thanks, this has been changed.

Exclusion of lines 43- 46: “diagnosis of pregestational nutrition”

Inclusion of lines 43- 44: “the pregestational nutritional status”

Materials & Methods

Pre-pregnancy weight was self-reported correct? If data has been extracted from the medical record, why not use that pre-pregnancy weight measurement?

Replay: Patients are referred to the clinic only when they are pregnant; there are no previous medical records available. This is why we opted for the self-reported prepregnancy weight.

Were the surveys completed at the doctor’s office during the prenatal visit? Were they conducted on paper or online? A bit more explanation of these procedures would be appreciated.

Reply: Thanks, I clarified this in lines 66-74 as follows:  

“After the diagnosis of GDM, all of the women were invited to participate in a multi-disciplinary group to receive guidance on diet, PA, and glucose monitoring. If the treatment goal (fasting glucose < 95mg/dL and 1h postprandial <140mg/dL) was not reached with lifestyle changes, insulin therapy was indicated. Before joining the mul-tidisciplinary group, they were invited to participate in the research protocol of this study. Those who agreed to participate received all the information concerning the study. Only after reading and signing the FICS, were they asked about their pregesta-tional weight, food consumption, and PA on the preestablished structured question-naires on paper [13].”

Figure 1: There are two adequate weight gain boxes – should one be insufficient weight gain?

Reply: Thanks, it was corrected. 

What was the mean gestational age? Was there a cut off for gestational age and recruitment?

GDM diagnosis timing would be an interesting statistic to know as well.

Reply: Gestational age was not an inclusion criterion, and there was no significant statistical difference between the groups regarding gestational age at diagnosis, at the beginning of follow-up, and at the time of delivery.

The following was included in lines 151-155:

“Gestational age (GA) at GDM diagnosis (EWG 16,30 ± 8,42 vs. AWG 16,24 ± 8,89 and IWG 17,68 ± 8,18; p = 0,647), GA at the first evaluation of GDM (EWG 23,16 ± 7,33 vs. AWG 23,21 ± 7,83 and IWG 23,14 ± 6,89; p= 0,982) and GA at delivery (EWG 38.70 ± 1.11 vs. AWG 38.03 ± 1.88 and IWG 38.48 ± 1.27; p = 0.093) did not differ statistically between the groups.”

Since there was a small percentage of primiparous women, did GWG differ between primi and others? These results would be interesting especially if there was a higher incidence of GDM in women with previous pregnancies.

Reply:

No there was no difference in the distribution of primigravida and personal history of GDM between groups. (Table 1)          

Primigravida p= 0.670

No

41 (71.9)

37 (78.7)

49 (72.1)

Yes

16 (28.1)

10 (21.3)

19 (27.9)

Personal history of GDM p= 1.00

No

52 (91.2)

43 (91.5)

61 (89.7)

Yes

5 (8.8)

4 (8.5)

7 (10.3)

Lines 138 and 165: Are these the correct abbreviations? In the abstract, adequate weight gain is AWG and insufficient weight gain is IWG and these are how they are labeled in the tables.

Reply: Thanks, they were corrected.

Line 185: Bold MET

Reply: Thanks, it was corrected.

Insulin treatment seemed to be the main focus of these results…were there other treatment options women? It would be interesting to see how these treatment options influenced dietary patterns and physical activity especially since those two are main treatment options for GDM.

Reply: In the first multidisciplinary group meeting, when the first data were collected, pregnant women received instructions on diet, PA, and glucose monitoring. Medical advice was initially given, clarifying queries and questions about GDM. Subsequently, a nutritionist provided guidelines for the best diet to be followed by a pregnant woman with GDM. In relation to self-monitoring of blood glucose, a nurse provided instruction and learning opportunities for the pregnant women to use the glucose meter, and they started to perform 4 capillary blood glucose measurements per day. If treatment goals were not achieved (fasting ≤ 95 mg / dL and 1 h postprandial ≤ 140 mg / dL) with lifestyle changes in 70% of capillary blood glucose measurements assessed in one to two weeks, insulin therapy was indicated and the patient started to monitor capillary blood glucose 6 or 7 times a day.

It has been included in lines 66-74.

Line 246: Are these pregnant specific recommendations or for the general public?

The paragraphs starting at line 244-248 and 249-257 – can you connect these points better? The first paragraph talks about METs/week but the second paragraph talks about 150 min/week. Having those connect on one type of outcome will create a stronger point and be more cohesive.

Reply: These recommendations are for the general public, but the 2019 Canadian Guideline for Physical Activity throughout Pregnancy recommends at least 150 min of moderate-intensity physical activity each week, which corresponds to 450 to 750 METs/week.

Inclusion of lines 265-270: According to Haskell et al. [30], for adults to maintain their health, their individual energy expenditure should lie between 450 to 750 METs/week after adding up all of the physical activities undertaken, be they sports or daily movements. The specific recommendations for physical activity throughout pregnancy should equal at least 150 min of accumulated moderate-intensity PAs per week (which corresponds to 450 to 750 METs/week), to achieve clinically meaningful health benefits and reductions in pregnancy complications [31].

Inclusion of reference:

  1. 3 Mottola, M.F.; Davenport, M.H.; Ruchat, S.M.; Davies, G.A.; Poitras, V.; Gray, C.; Jaramillo, A.; Barrowman, N.; Adamo, K.B.; Duggan, M.; et al. No. 367-2019 Canadian Guideline for Physical Activity throughout Pregnancy. J. Obstet. Gynaecol. Canada 2018, 40, 1528–1537, doi:10.1016/j.jogc.2018.07.001.

If you need any further information, please do not hesitate to contact us.

Sincerely,

Rossana Pulcineli Vieira Francisco

Reviewer 2 Report

Due to the growing incidence of pregnancy diabetes (GDM), as well as early and late GDM disorders, concerning both the mother and the child, I consider the subject to be right and justified. 

The research conducted in this area is both cognitive and, above all, application-related, which has been shown in the introduction, as well as an extensive discussion prepared on the basis of 39 items of literature from the last decade.

The design of the study is typical for original studies. Research methods have been properly selected and described, although with regard to the selection of the research group, a complete list of exclusion factors should be presented, and in the clinical and demografic data part, the frequency of measuring body mass indicators in context of gestation age should be provided.

The research results were properly prepared and presented both in descriptive form, as well as in the form of 1 figure and 3 tables. In the case of the above-mentioned figure, I notice an editorial error. In scope of the final sample of GDM, the indication of the AWG group was repeated twice and the IWG was misssing.  Methods of statistical analysis of the results deserve a positive assessment application of multinominal logistic regression analysis, Conclusions are clearly related to the results and indicate the full achievement of the purpose study.

The part referred to as study limitation proves the scientific maturity of the authors.  It would be worth mentioning the use of the results obtained in a non-Brazilian pregnancy population (if available) as well as the need for similar studies in pregnant women who did not develop gestation diabetes (control group). In this respect the interpretation could also be broadened.

Author Response

May 13, 2021

Dear Reviewer,

Thank you for your review. Following the corrections explaining point-by-point the details of the revisions are our responses as requested.

Due to the growing incidence of pregnancy diabetes (GDM), as well as early and late GDM disorders, concerning both the mother and the child, I consider the subject to be right and justified. 

The research conducted in this area is both cognitive and, above all, application-related, which has been shown in the introduction, as well as an extensive discussion prepared on the basis of 39 items of literature from the last decade.

The design of the study is typical for original studies. 

Reply: Thanks for your considerations.

Research methods have been properly selected and described, although with regard to the selection of the research group, a complete list of exclusion factors should be presented,

Reply: Thanks, I clarified this in lines 55-64 as follows:  

“The study included pregnant women aged 18 years or older with a single fetus, a diagnosis of GDM according to the International Association of Diabetes and Pregnancy Study Groups [12] (initial fasting glucose ≥ 92 mg / dl and <126 mg / dl or 75 g oral glucose tolerance test with fasting ≥ 92 mg/dl and / or after 1 h ≥ 180 mg/dl and / or after 2 h ≥ 153 mg/dl), absence of glucose intolerance prior to pregnancy (defined by previous diagnosis of polycystic ovary syndrome or by fasting glucose ≥ 100 mg / dl or 2 h after overload with 75 g of glucose ≥ 140 mg / dL or glycated hemoglobin ≥ 5.7% before pregnancy), nonchronic use of glucocorticoids or antiretroviral drugs for HIV viruses, given their diabetogenic effect and potential confounding factor, and a signed free and informed consent statement (FICS). Excluded were the pregnant women who could not understand and/or respond to the research questionnaire items, who missed the prenatal care visits, and whose medical records were incomplete.”

In the clinical and demografic data part, the frequency of measuring body mass indicators in context of gestation age should be provided.

Reply: Thanks, It has been clarified in lines 85-88 as follows:  

“The pregnant women’s weight was measured in every medical consultation; however, in this study we used only the last weight measurement, which was made at the time of delivery, to calculate the total weight gain during pregnancy.”

The research results were properly prepared and presented both in descriptive form, as well as in the form of 1 figure and 3 tables. In the case of the above-mentioned figure, I notice an editorial error. In scope of the final sample of GDM, the indication of the AWG group was repeated twice and the IWG was misssing. 

Reply: Thanks, the editorial error was corrected. 

Methods of statistical analysis of the results deserve a positive assessment application of multinominal logistic regression analysis, Conclusions are clearly related to the results and indicate the full achievement of the purpose study.

The part referred to as study limitation proves the scientific maturity of the authors. 

It would be worth mentioning the use of the results obtained in a non-Brazilian pregnancy population (if available) as well as the need for similar studies in pregnant women who did not develop gestation diabetes (control group). In this respect the interpretation could also be broadened.

Reply: Thank you for your considerations. We included the following paragraph at the end of the Conclusions. Lines 336- 338:

“Also, additional research should be carried out to clarify the relationship between diet quality and eating patterns before and during pregnancy in the Brazilian and non-Brazilian populations.”

If you need any further information, please do not hesitate contact us.

Sincerely,

Rossana Pulcineli Vieira Francisco
